# Novel Treatment for Pre-XDR Tuberculosis Linked to a Lethal Case of Acute Myocarditis

**DOI:** 10.3390/diagnostics14192139

**Published:** 2024-09-26

**Authors:** Serafeim-Chrysovalantis Kotoulas, Pavlos Poulios, Georgia Chasapidou, Elena Angeloudi, Triantafyllenia Bargiota, Maria Stougianni, Katerina Manika, Eleni Mouloudi

**Affiliations:** 1Adult ICU, General Hospital of Thessaloniki “Hippokration”, National Healthcare System, 54642 Thessaloniki, Greece; akiskotoulas@hotmail.com (S.-C.K.); pgpoulios@gmail.com (P.P.); fillbargiota@hotmail.com (T.B.); mstougianni@gmail.com (M.S.); elmoulou@yahoo.gr (E.M.); 2Pulmonary Department, General Hospital of Thessaloniki “G. Papanikolaou”, National Healthcare System, 57010 Thessaloniki, Greece; giorgia_cha@yahoo.gr; 3Adult CF Unit, Pulmonary Department, General Hospital of Thessaloniki “G. Papanikolaou”, Aristotle’s University of Thessaloniki, 57010 Thessaloniki, Greece; ktmn05@yahoo.gr

**Keywords:** adverse event, drug-induced myocarditis, pre-extensively drug-resistant tuberculosis, pretomanid, case report, low ejection fraction

## Abstract

The management of resistant tuberculosis (tb) can be extremely difficult, especially in case of novel unpredicted complications. In this report, we present a case of a 48-year-old patient with pre-extensively drug-resistant (XDR) tb who received a treatment regimen including pretomanid, bedaquiline, linezolid, cycloserine, and amikacin and died due to myocarditis. Acquired resistance to first- and second-line drugs developed due to previous poor adherence to medication. The clinical presentation of the patient, along with her initial ultrasonographical, electrocardiogram (ECG), and laboratory examinations, were typical for acute myocarditis; however, the patient was considered unstable, and further investigations, including magnetic resonance imaging (MRI), pericardiocentesis, and endomyocardial biopsy were not performed. To our knowledge, this is the first case of myocarditis in such a patient, the clinical features of which raised a high suspicion of drug induction that could be attributed to the treatment regimen that was administered. Clinicians who manage cases of drug-resistant tb should be aware of this newly reported, potentially lethal, adverse event.

## 1. Introduction

The management of patients with pre-extensively drug-resistant tuberculosis (pre-XDR-tb) is particularly demanding. Tb treatment relies on a multi-drug therapy over a long duration, and as a result, the emergence of adverse reactions is inevitable [1]. Novel anti-tuberculous (anti-tb) drugs include combinations with fluoroquinolones, aminoglycosides, bedaquiline, pretomanid, linezolid, cycloserine, linezolid, and clofazimine [2]. Minor adverse effects are relatively common, and they can be easily managed with symptomatic treatment. However, new unexpected challenges can always emerge in everyday practice, especially in cases of potentially lethal adverse events, which might be related to those novel drugs.

Several anti-tb drugs may be associated with significant cardiac toxicity; bedaquiline, clofazimine, delamanid, and fluoroquinolones (especially moxifloxacin) have been associated with QTc prolongation, leading to cardiac arrhythmia and/or death. If the QTc > 500 ms, the suspected drug(s) should be stopped, electrolytes should be checked and corrected if needed, and electrocardiograms should be monitored until normalized. Risk factors for QTc prolongation need to be immediately detected and include patients with a history of torsade de pointes, congenital long QT syndrome, hypothyroidism, bradyarrhythmia, and uncompensated heart failure, as well as patients on other QTc-prolonging drugs [1]. In data submitted to the FDA, death was observed more frequently among patients receiving bedaquiline than among those received other anti-tb drugs. Clofazimine and bedaquiline have a synergic effect and together are more likely to cause QTc prolongation, especially if they are additionally combined with other QTc-prolonging drugs [3]. However, there are no reports regarding an association between myocarditis and the use of novel anti-tb drugs. To our knowledge, this is the first case of myocarditis in a patient with a history of pre-XDR-tb, the clinical features of which raise a high suspicion of drug induction that could be attributed to the treatment regimen that was administered.

## 2. Case Report

A 48-year-old Eastern European female with a relapse of cavitary pulmonary pre-XDR-tb, without extrapulmonary manifestations, for which she had previously received first- and second-line drugs without success due to non-adherence to the treatment regimens, was admitted to our hospital and an in-hospital treatment regimen was initiated. Based on the 2022 WHO guidelines [1], a six-month treatment regimen composed of bedaquiline, pretomanid, linezolid (600 mg), and moxifloxacin (BPaLM) is suggested for MDR/RR-tb patients. Quinolones were excluded from the regimen since the strain isolated from the patient was ofloxacin-resistant and the patient had unfortunately received moxifloxacin as a monotherapy for extended periods of time in the past. Resistance to linezolid, which the patient had also previously received, could not be determined. On this basis an extended off-label regimen consisting of pretomanid, bedaquiline (to which the patient was naive) linezolid, cycloserine, and amikacin was selected. Cycloserine and amikacin were active based on the DST and were added to the regimen because of the uncertain activity of linezolid, in order to strengthen the combination of bedaquiline and pretomanid.

Regarding previous treatment regimens, the patient had received the first line treatment (isoniazid, rifampicin, ethambutol, pyrazinamide) between October 2008 and March 2009 and again between June 2009 and November 2009 and the second line treatment (moxifloxacin, amikacin, para-aminosalicylic acid, pyrazinamide, ethionamide, cycloserine, capreomycin) between November 2010 and April 2012, while between the two regimens she also received isoniazid, rifampicin, pyrazinamide, and moxifloxacin for five months. All these treatment regimens were given to the patient by her doctors in Eastern Europe. She came to our clinic and initiated the last treatment regimen in November 2022.

Upon admission, her clinical status was poor due to her extensive pulmonary involvement (Figure 1). Her oxygenation was marginal (SaO_2_: 92%), she presented with cough, purulent sputum, hemoptysis, fever, night sweats, and considerable weight loss (body mass index (BMI): 17.8 Kg/m^2^). Her sputum smear and culture were positive for pre-XDR-tb. Her history was otherwise unremarkable.

After following the aforementioned treatment regimen for 35 days, her clinical status improved, as her symptoms declined, she gained weight, and her smear sample, obtained via bronchoscopy, converted to negative. Despite her physicians’ advice, she was discharged from hospital on her own will, and continued her treatment at home. Five days later, she presented to the emergency department with worsening dyspnea from three days. She was hemodynamically unstable despite fluid resuscitation, while her oxygenation was severely low (pO_2_: at 37 mmHg at room air) and did not improve sufficiently after initiating treatment with high flow nasal cannula (HFNC); thus, she was intubated and admitted to the intensive care unit (ICU).

In the ICU, a cardiac ultrasound revealed the following images and videos (Figure 2, Appendix A).

The cardiac ultrasound was remarkable for severe systolic dysfunction of the left ventricle (LV) with akinesia of the mid and apical segments (Appendix A) compensatory basal hyperkinesis (Appendix A), and a total ejection fraction (EF) of <20% (Appendix A). Pericardial fluid was also evident, mainly in front of the right ventricle (~1.9 cm), without signs of incipient tamponade (Figure 2).

Laboratory testing revealed an increase in hs-troponin (1655 pg/mL, range < 11.6 pg/mL), SGOT (638 U/L), and SGPT (358 U/L). The electrocardiogram (ECG) upon admission was remarkable for negative T waves in I, aVL, V5, and V6 leads and a poor R progression in the precordial leads, without QT interval prolongation (Figure 3).

Based on the patient’s clinical presentation, along with her ECG, cardiac ultrasound, and laboratory tests, the primary diagnosis that was crucial to confirm or exclude was acute myocardial infarction. In this case, a coronary computer tomography (CT) angiography was performed, from which no critical stenosis was found (calcium score = 0). A coronary angiography, though an invasive method compared to coronary CT angiography, might be more appropriate in such cases, since it provides the ability to treat a critical stenosis, if any, in the coronary arteries [4], even though in this case it was proved that the problem was not as such.

A massive pulmonary embolus could cause severe hypoxia along with hemodynamic instability and match with the clinical presentation. The patient’s laboratory findings and some of the ECG findings might also be present in a massive pulmonary embolism; however, a cardiac ultrasound would have revealed different findings, such as an enlarged right ventricle (RV), possibly larger than the LV, with or without a D-shaped intraventricular septum, along with signs of RV dysfunction, such as a decreased tricuspid annular plane systolic excursion (TAPSE) and possibly an increased pulmonary artery systolic pressure (PASP) [5], making the diagnosis of a massive pulmonary embolism unlikely and the performance of a CTPA unnecessary.

The patient might also have suffered from pulmonary hypertension due to her chronic pulmonary disease; however, pulmonary hypertension does not usually present with such a fulminant insult [6]; thus, it would be rather unlikely to be the cause of the patient’s clinical presentation. On the other hand, continuous hemodynamic monitoring would be extremely helpful in this case; however, other, less invasive methods than Right Heart Catheterization (RHC) are available for that [7]; thus, at this stage, RHC might have been not only unnecessary, but also risky, since, as an invasive modality, it is associated with various complications [8]. In this case, minimally invasive continuous hemodynamic monitoring confirmed the low cardiac output (CO) (cardiac index (CI):1.4 L/min/m^2^).

Due to the low CO and the clinical signs of tissue hypoperfusion, treatment with dobutamine and phenylephrine was initiated, along with colchicine for the pericardial effusion. In the following days, the patient’s LV systolic function was restored and her pericardial fluid diminished. Her high-sensitivity (hs-troponin) levels also decreased and remained stable between 100 and 200 pg/mL for the remainder of her hospitalization. She also presented eosinophilia (>1000/μL) between the 5th and 9th day after her admission, which was not related with drug initiation or discontinuation or stool parasites, while her complete serological examinations for infectious and auto-immune diseases came back negative. Her follow-up ECGs were notable for a low atrial rhythm, ST-segment repolarization disorders, and T-wave inversions of varying amplitudes, which progressed through different stages over time (Figure 4, Figure 5, Figure 6, Figure 7 and Figure 8).

Despite her cardiac function restoration, the patient was unable to wean from mechanical ventilation due to the severe damage to her lung parenchyma and eventually died four weeks later from septic shock due to Klebsiella pneumonia.

## 3. Discussion

The clinical presentation of the patient, along with her initial ultrasonographical, ECG, and laboratory examinations and their progression throughout the course of her hospitalization, were typical for acute myocarditis [9], although the diagnosis and its etiology could not be investigated further with cardiac magnetic resonance imaging (MRI) or pericardiocentesis, as the patient was considered unstable and her pericardial fluid was relatively low in quantity. Due to her critical condition, a cardiac biopsy was also not performed.

An obvious cause of the patient’s condition was tb of the myocardium and pericardium. The anti-gravity localization of the pericardial fluid might suggest a tb pericardial effusion with adhesion that caused loculation. However, this possibility seems rather unlikely since the patient reported the resolution of her symptoms and had responded positively to treatment. Furthermore, tb myocarditis is rare (prevalence varies between 0.14% and 2%) and also presents and progresses insidiously rather than acutely [10,11].

In a recent expert consensus document [12], the causes of myocarditis were grouped in three categories: (1) infectious, (2) immunological, and (3) drug-induced [12]. In this case, the patient had no history of systemic inflammatory disease, and her serological examinations for both auto-immune and infectious diseases were negative, leading to the possibility of drug reaction. Drug-induced myocarditis is characterized by eosinophilic infiltration of myocardium and peripheral eosinophilia, which might be evident in the course of the disease, but can be absent in up to 25% of the cases at admission [12,13,14]. This type of drug hypersensitivity reaction is particularly difficult to recognize because clinical features such as fever, malaise, and skin rash might be absent in most cases [14,15,16]. The latency period from drug initiation to the emergence of myocarditis typically varies between 2 and 5 weeks for the majority of the implicated substances [14,15,16,17,18,19]. Although our patient’s characteristics match with all the aforementioned clinical features, unfortunately, the diagnosis could not be confirmed with cardiac MRI, pericardiocentesis, or cardiac biopsy.

Drug-induced acute myocarditis seems to be the most probable diagnosis in this case; however, not only could the etiology of the myocarditis not be proved with certainty, but also none of the drugs that the patient received has been implicated with such a diagnosis in the past. Amikacin is associated with nephrotoxicity and ototoxicity, and linezolid is associated with bone marrow suppression (in this case, the patient had also received linezolid in the past), while cycloserine is linked to psychotic symptomatology. Although antipsychotics are the most frequently related drugs to acute myocarditis [14], the patient was under supervision and had not received any drugs from this category while being treated in the hospital for her pre-XDR tb prior to her ICU admission.

Nevertheless, the patient did receive recently approved anti-tb medications, which alone, or co-administered with other drugs, might have triggered this reaction. More particularly, even though bedaquiline has been in circulation for quite a while, pretomanid is a novel anti-tb medication, which, up to early 2021, has been administered in varying dosing regimens, alone, or in combination with other anti-tb drugs, in eight clinical trials with 1168 participants in total [20], and although myocarditis has not been reported as an adverse event [20], the number of patients that have been treated with it is too small to safely exclude this case scenario.

Last but not least, after reporting the suspected adverse events of anti-tb drugs, another important issue is to establish whether there is a causal or temporal relationship between the drug and the adverse event. In other words, it is possible that the administered drug and the manifestation of an adverse event may have a close chronological relationship but still do not represent a reaction. Pharmacovigilance dictates that clinicians who manage cases of pre-XDR-tb should be aware of such a contingency.

## 4. Conclusions

In conclusion, the management of pre-XDR-tb is particularly demanding [1], and novel anti-tb drugs have been proved to be extremely helpful [2]; however, new unexpected challenges can always emerge in everyday practice, especially in cases of potentially lethal adverse events, which might be related to those novel drugs. This is the first study to report a lethal case of myocarditis that could be attributed to the administration of novel anti-tb drugs alone, or combined with other medications, for the treatment of pre-XDR-tb.

## Figures and Tables

**Figure 1 diagnostics-14-02139-f001:**
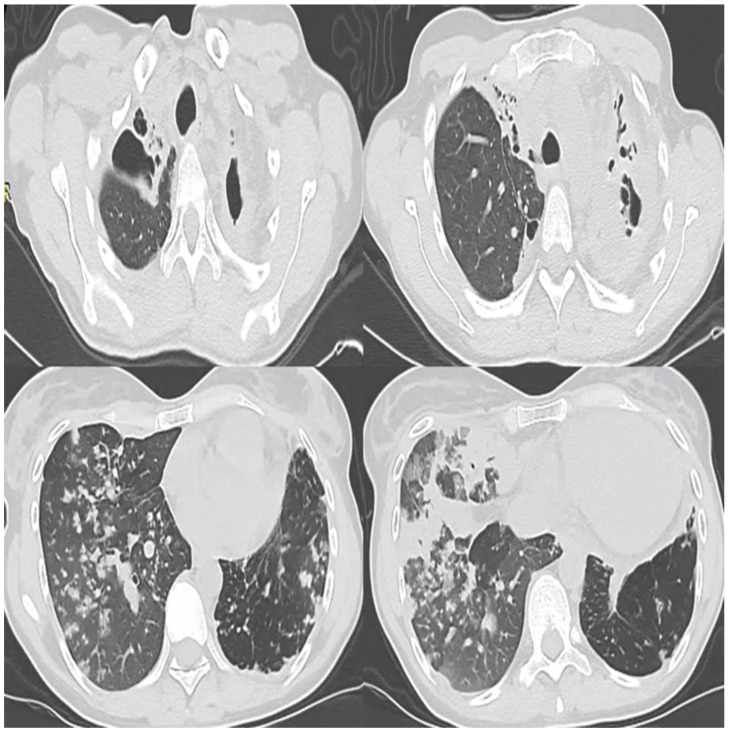
Bilateral extensive pulmonary involvement. Bilateral upper lobe cavities with consolidation. Middle lobe consolidation. Bilateral lower lobe infiltrates with tree in bud appearance.

**Figure 2 diagnostics-14-02139-f002:**
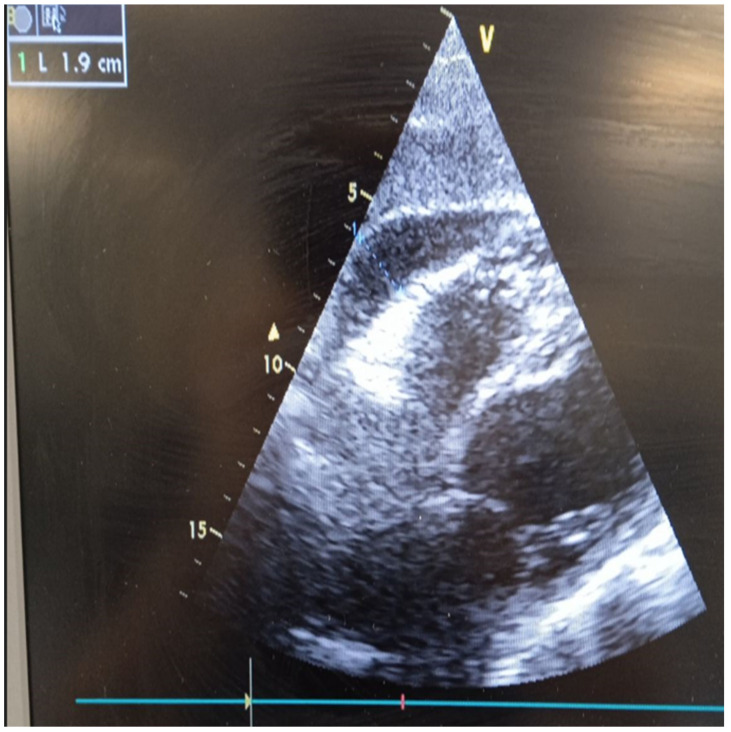
Cardiac ultrasound at systolic phase showing pericardial effusion in front of the right ventricle (~1.9 cm), without signs of incipient tamponade.

**Figure 3 diagnostics-14-02139-f003:**
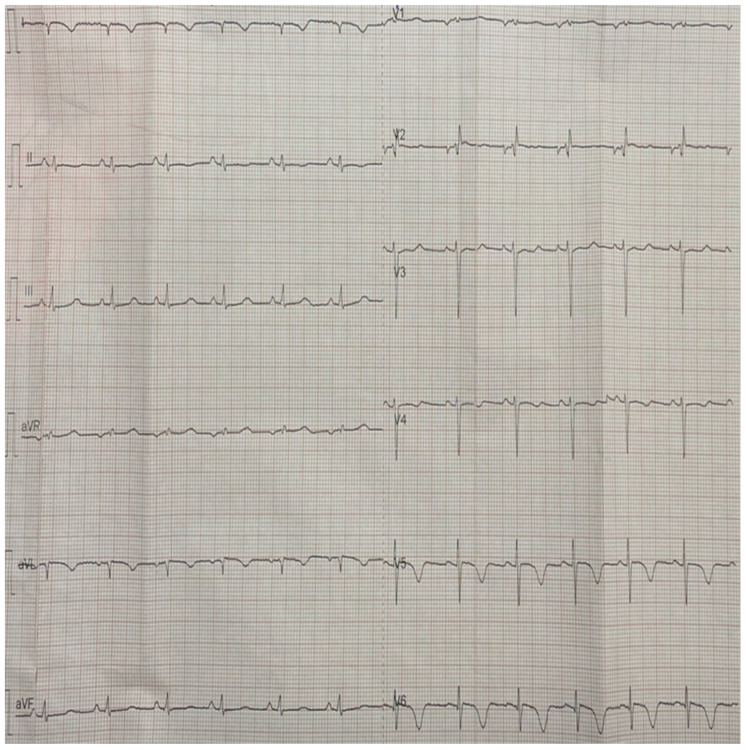
ECG findings from day 1 (upon admission) showing negative T waves in I, aVL, V5, and V6 leads and a poor R progression in the precordial leads. These findings are suggestive for Regional Wall Motion Abnormalities (RWMAs) towards the lateral wall.

**Figure 4 diagnostics-14-02139-f004:**
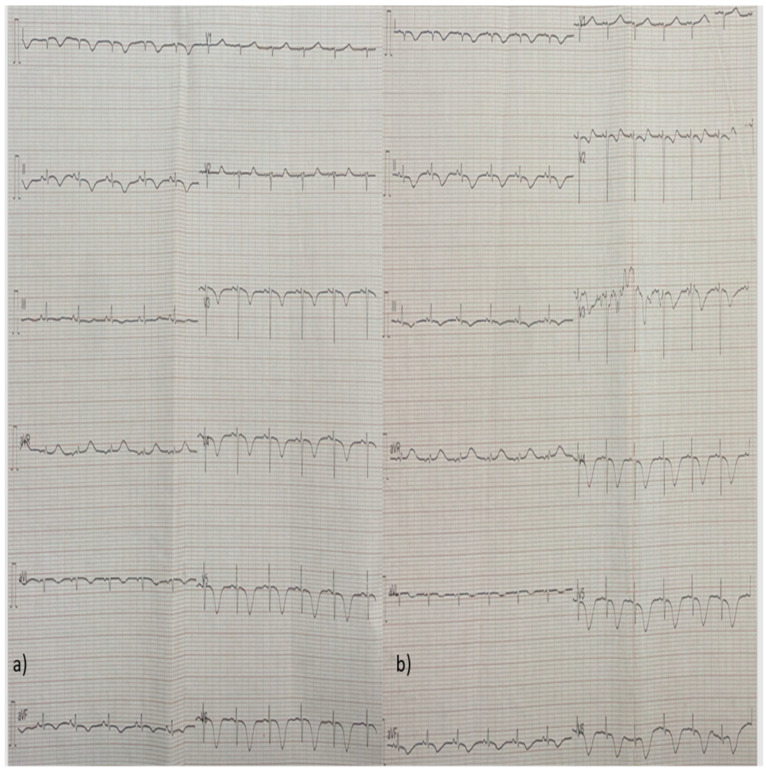
(**a**,**b**) Follow-up ECG from the next two consecutive days (day 2 and day 3) showing ST-segment repolarization disorders and T-wave inversions expanding towards the anterior and inferior wall, apart from the lateral wall.

**Figure 5 diagnostics-14-02139-f005:**
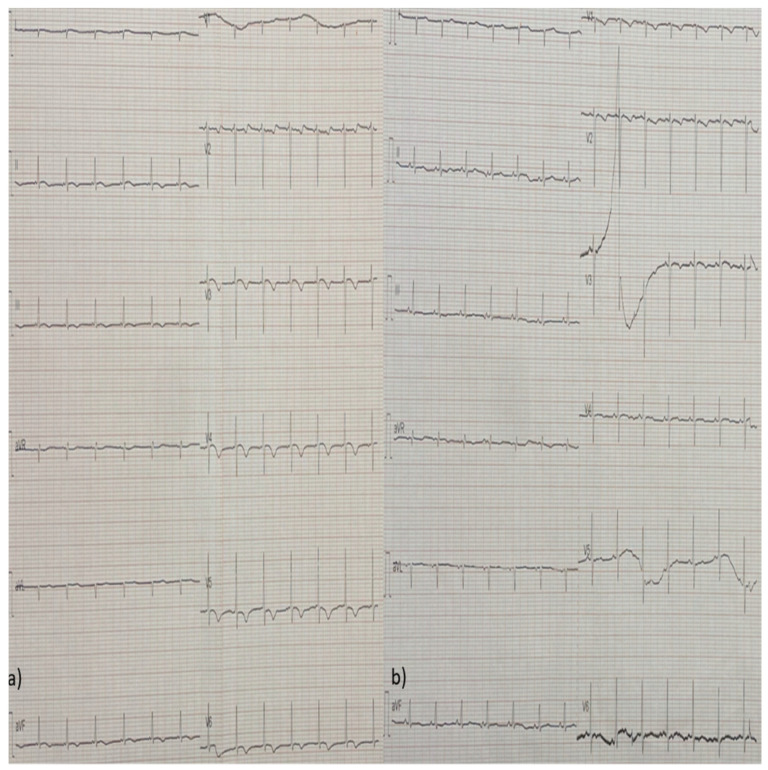
(**a**,**b**) Follow-up ECGs (day 8 and 9) showing improvement in ST-segment repolarization disorders and T-wave inversions observed during the previous days, especially in the inferior wall.

**Figure 6 diagnostics-14-02139-f006:**
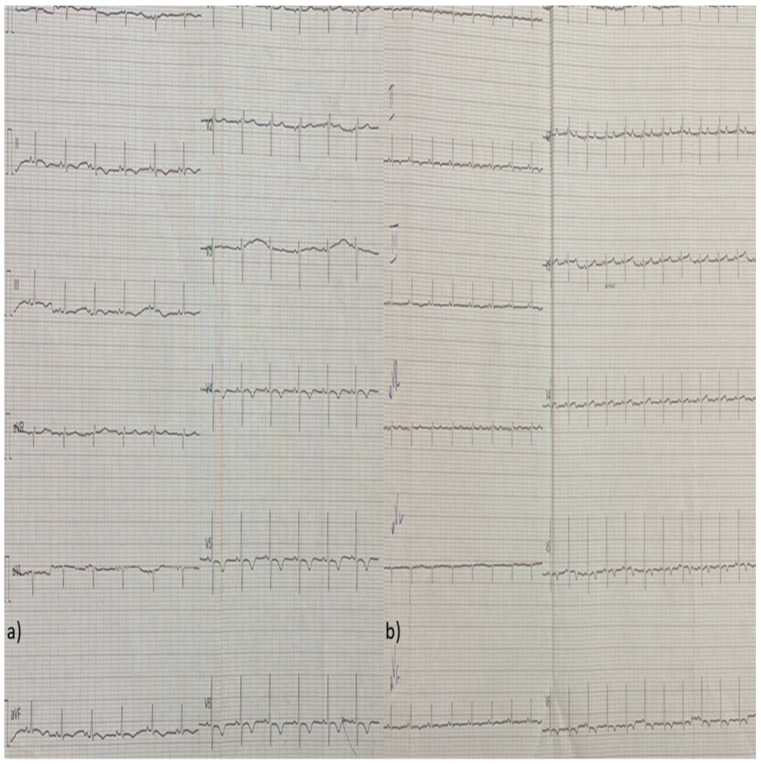
(**a**,**b**) Follow-up ECGs from day 12 and 13 showing that the aforementioned improvement in T-wave inversions, described in the previous figures, has progressed to the anterior and lateral walls.

**Figure 7 diagnostics-14-02139-f007:**
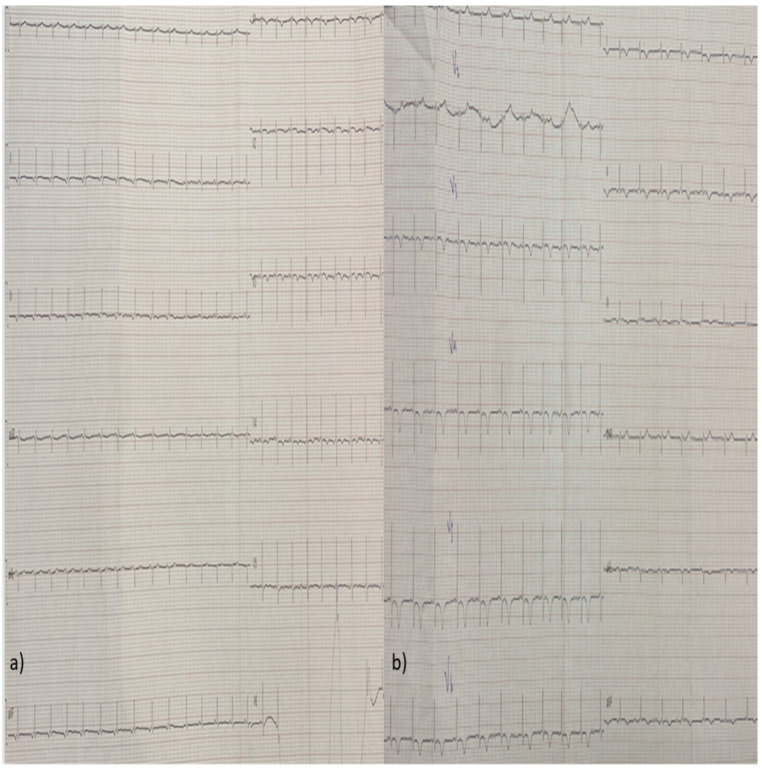
(**a**,**b**) On follow-up ECG (day 15, **a**) there is improvement in T-wave inversions, as the ECG describes repolarization abnormalities concerning the affected walls. However, there is an obvious reoccurrence of T-wave inversion on day 17 (**b**), depicting the dynamic character of myocarditis ECG abnormalities, in contrast with acute coronary syndrome.

**Figure 8 diagnostics-14-02139-f008:**
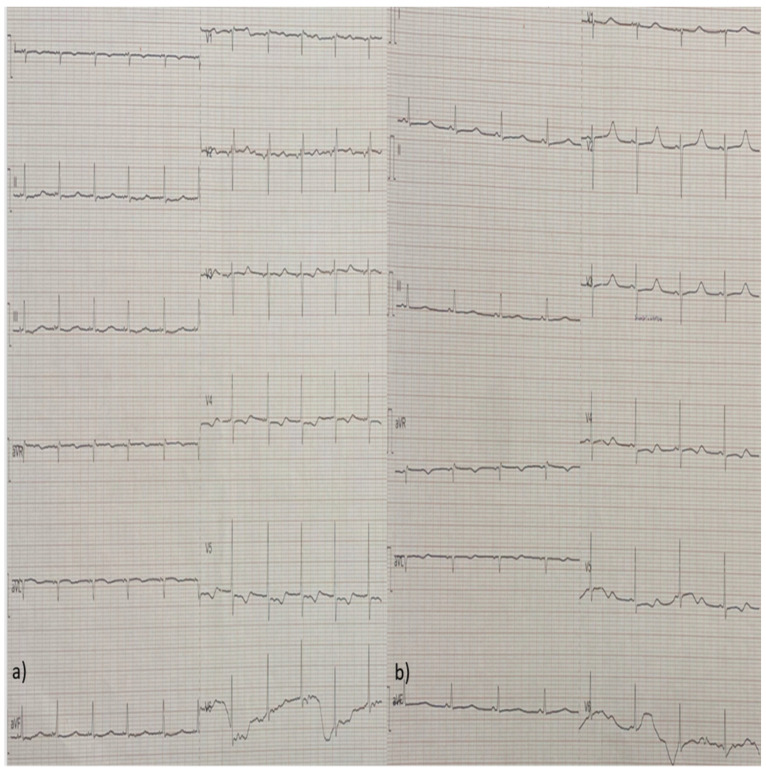
(**a**,**b**) Follow-up ECG on day 19 (**a**) showing repolarization abnormalities (mild ST-segment depression) of the affected walls. Follow-up ECG on day 21 (**b**) showing long QT for the first time along with mild repolarization abnormalities.

## Data Availability

The datasets used and/or analyzed during the current study are available from the corresponding author on reasonable request.

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
