# Peer review of "Novel Treatment for Pre-XDR Tuberculosis Linked to a Lethal Case of Acute Myocarditis"

_diagnostics, 2024, doi:10.3390/diagnostics14192139_

Round 1

Reviewer 1 Report

Comments and Suggestions for Authors

Congratulations to the team of authors, it is really necessary to track adverse reactions in patients who receive associated antituberculosis medication and the exchange of experience brings progress.

I have the following observations. In the introduction, it would be useful to know which treatment recommendation from the national guide or if they use international references.

R 40-45. - I would detail the type of adverse cardiac reactions, because there are several described in the literature, at least for bedaquiline.

Some data on the laboratory diagnostic modality would be welcome, for example liquid cultures, resistance testing how was it done, was genetic diagnosis used. Is it a chronic case? What is the resistance spectrum of the detected MT.

What is the ECG evaluation at the initiation of treatment, which drugs were administered in previous regimens?

Are biological parameters changed at the time of hospitalization or discharge?

All ECGs must be interpreted identically, specifying in the description the moment of the test, which are the derivations that have changes, respectively which cardiac territory corresponds to the derivation (inferior, lateral, anterior) compared to the previous evaluation.

The case is extremely interesting and I encourage the team to bring some improvements to the current form to be able to offer the experts their clinical experience.

Author Response

1) I have the following observations. In the introduction, it would be useful to know which treatment recommendation from the national guide or if they use international references.

Thank you for your excellent comment. We used the 2022 WHO guidelines and we have made the respective changes in the 1st paragraph of the case description of our revised manuscript.

2) R 40-45. - I would detail the type of adverse cardiac reactions, because there are several described in the literature, at least for bedaquiline.

Thank you for pointing that out. We have provided more details about the adverse cardiac reactions of Bedaquiline in our revised manuscript.

  • Some data on the laboratory diagnostic modality would be welcome, for example liquid cultures, resistance testing how was it done, was genetic diagnosis used. Is it a chronic case? What is the resistance spectrum of the detected MT.

Thank you for your comment. The patient was initially diagnosed with TB in 2008 in Eastern Europe, Georgia. However, she has not completed a specific treatment regimen due to non-compliance. We have available culture and genetic diagnosis results from 2022. Ziehl-Neelsen stain was negative. Löwenstein–Jensen sputum culture was positive for Mycobacterium and showed sensitivity to Clarythromycin, Capreomycin, Amikacin, Cycloserine and Ethionamide. MGIT liquid culture system showed sensitivity to PZA. Mycobacterium was possibly resistant to Kanamycin. Molecular identification using rRNA MTB (TOSOH) procedure showed that the strain belongs to the genus Mycobacterium. DNA MTBC PCR (XPERT) detected rifampicin resistance mutation.

4) What is the ECG evaluation at the initiation of treatment, which drugs were administered in previous regimens?

Thank you for your constructive points. We have addressed all your points in our revised manuscript (lines 74-82). The patient had received the 1st line treatment (HRZE) between 10/2008 and 03/2009 and again between 06/2009 and 11/2009 and the 2nd line treatment (Moxifloxacin, Amikacin, PAS, Z, Ethionamide, Cycloserine, Capreomycin) between 11/2010 and 04/2012, while between the two regimens she also received HRZ + Moxifloxacin for 5 months. All these treatment regimens were given to the patient by her doctors in Eastern Europe. She came in our clinic and initiated the last treatment regimen at 11/2022.

Baseline ECG evaluation at the initiation of treatment was normal.

5) Are biological parameters changed at the time of hospitalization or discharge?

Thank you for your constructive comment. The biological parameters of the patient during first hospitalization improved and the patient was discharged (on her own will) afebrile, hemodynamically stable with adequate oxygenation to continue her treatment at home without need for supplementary oxygen therapy.

6) All ECGs must be interpreted identically, specifying in the description the moment of the test, which are the derivations that have changes, respectively which cardiac territory corresponds to the derivation (inferior, lateral, anterior) compared to the previous evaluation.

We appreciate the time and effort that you dedicated to providing your valuable comment. We agree that a detailed interpretation of the ECGs was missing, and therefore, we have revised the legends of the figures accordingly.

Reviewer 2 Report

Comments and Suggestions for Authors

This is a fatal case of non-compliant pre-XDR tuberculosis complicated with acute myocarditis after treatment with the second-line anti-TB regimen. The patient died of ventilator-associated pneumonia (VAP). The probable cause of myocarditis in this case is unknown, and the authors have raised some speculation. Myocarditis is rare but not unheard of in TB, with an estimated incidence of <0.2% [Fairley CK et al. J Infect. 1996; 32: 223-225.] Cardiac involvement in TB is more common than we previously believed if clinicians are willing to investigate more aggressively [reference 9]. Although several drugs prescribed to the patient have been reported to be associated with prolonged Q-Tc, frank myocarditis due to anti-TB medications is difficult to find in the literature. The still frame of the echocardiogram is not informative, except for the evidence of pericardial effusion, without an EKG waveform to tell which phase the image was taken. A M-mode might be more helpful. It is also curious why the pericardial effusion was located on the anti-gravity side of the heart instead of the retrocardiac side. I agree that the supplemental video shows poor contractility of LV without evidence of RVH, judging from the RV size and IVS excursion. Still, the pericardial effusion is not too concerning to a physician who works in an intensive care unit like me. The reason why the RV is so hyperechogenic was not explained. There are six EKGs, which can be arranged into one picture for easy comparison. Interestingly, the QT was not prolonged throughout the course, while prolonged QTc is the most common finding in acute myocarditis [Lazzerini PE. et al. Front Cardiovasc Med. 2015 May 27;2:26.] Is it possible that the myocarditis is due to severe hypoxemia [Wang JG et al. Crit Care Explor. 2021 Feb 24;3(3):e0355.] as being reported in COVID-19? A few minor concerns are listed below.

-          Line 53:  “due to poor compliance to the treatment regimens” can be replaced by “non-compliance.”

-          It is unnecessary to capitalize tuberculosis unless it is the first word of a sentence.

-          Please spell out RHC. This is not a common acronym for readers from different specialties.  

-          Is the article also quoted under the references section?

-          The legend for Figure 1 should be more informative. Do not expect the readers to interpret it themselves.  

-          This is trivial, but was the written consent obtained from the patient as claimed, or from the family?

Comments on the Quality of English Language

It is not difficult to comprehend the manuscript, but a few places can be improved.

Author Response

1) This is a fatal case of non-compliant pre-XDR tuberculosis complicated with acute myocarditis after treatment with the second-line anti-TB regimen. The patient died of ventilator-associated pneumonia (VAP). The probable cause of myocarditis in this case is unknown, and the authors have raised some speculation. Myocarditis is rare but not unheard of in TB, with an estimated incidence of <0.2% [Fairley CK et al. J Infect. 1996; 32: 223-225.] Cardiac involvement in TB is more common than we previously believed if clinicians are willing to investigate more aggressively [reference 9]. Although several drugs prescribed to the patient have been reported to be associated with prolonged Q-Tc, frank myocarditis due to anti-TB medications is difficult to find in the literature. The still frame of the echocardiogram is not informative, except for the evidence of pericardial effusion, without an EKG waveform to tell which phase the image was taken. A M-mode might be more helpful. It is also curious why the pericardial effusion was located on the anti-gravity side of the heart instead of the retrocardiac side. I agree that the supplemental video shows poor contractility of LV without evidence of RVH, judging from the RV size and IVS excursion. Still, the pericardial effusion is not too concerning to a physician who works in an intensive care unit like me. The reason why the RV is so hyperechogenic was not explained. There are six EKGs, which can be arranged into one picture for easy comparison. Interestingly, the QT was not prolonged throughout the course, while prolonged QTc is the most common finding in acute myocarditis [Lazzerini PE. et al. Front Cardiovasc Med. 2015 May 27;2:26.] Is it possible that the myocarditis is due to severe hypoxemia [Wang JG et al. Crit Care Explor. 2021 Feb 24;3(3):e0355.] as being reported in COVID-19? A few minor concerns are listed below.

Thank you for making all these constructive comments and give us the opportunity to improve this section of our manuscript. As far as the still frame of the echocardiogram, the ECG waveform was not available due to technical issues (one lead was disconnected, something that we had not noticed by that time), however, we considered that the phase of the cardiac cycle (diastolic phase) could be interpreted by the fact that mitral valve was closed. M-mode would definitely be more helpful, however, in the manuscript we used the image that we had available following our clinical assessment, which was diagnostic at that moment without the need to use M-mode.

Regarding your comment: “It is also curious why the pericardial effusion was located on the anti-gravity side of the heart instead of the retrocardiac side.”, this is an excellent observation, and we also do not have a definite answer for this phenomenon. We can only speculate that due to the decreased movement of the apical cardiac segments, a negative pressure gradient was generated into the pericardial sack, which caused the translocation of the pericardial effusion to the anti-gravity side of the heart.

We agree that the pericardial effusion was not too concerning, and therefore we did not perform a pericardiocentesis, a procedure which may be associated with serious complications. 

As far your comment “The reason why the RV is so hyperechogenic was not explained” is concerned, we believe that this is an artifact due to the high gain of the ultrasound settings.

Regarding your comment: “There are six EKGs, which can be arranged into one picture for easy comparison” we agree that it would be more appropriate, however, we tried to implement that and the quality of the figure was very poor, due to the compression of all 12 12-lead ECGs into one figure and thus, we decided not to change the format of the figures.

The most common ECG abnormalities seen in myocarditis include sinus tachycardia and non-specific ST segment and T waves changes, findings compatible with our case. Other ECG changes are variable, and may include: Prolonged QRS, QT prolongation, diffuse T wave inversion, ventricular arrhythmias as well as AV conduction defects. The ECG in some patients with myocarditis is similar to the ECG pattern of acute isolated pericarditis (which is suggestive of myopericarditis) or acute myocardial infraction (Butta2019, Diagnostic and prognostic role of electrocardiogram in acute myocarditis: A comprehensive review).

Common causes of myocarditis include viruses (coxsackie B virus, HIV, influenza A etc), bacteria (mycoplasma, rickettsia, leptospira), immunological causes (sarcoidosis, scleroderma, SLE etc), as well as drugs/toxins (including clozapine, amphetamines) (Reference 11). Indeed, as you accurately stated, possible causes of myocardial injury in patients with COVID-19 include hypoxic injury. However, in our case, we assume that hypoxemia occurred as a result of myocarditis, and not vice versa, since she was discharged with adequate oxygenation from her previous hospitalization 5 days earlier and new-onset hypoxemia was attributed to pulmonary edema caused by low ejection fraction of the left ventricle of the heart due to myocarditis. In other words, patient’s hypoxemia developed throughout the course of hospitalization and was not present upon admission.

2) Line 53: “due to poor compliance to the treatment regimens” can be replaced by “non-compliance.”

Thank you for pointing that out. We have made the respective change in the revised manuscript.

3)  It is unnecessary to capitalize tuberculosis unless it is the first word of a sentence.

Thank you for the comment. Following your recommendation, we have made the respective changes in the revised manuscript.

4)  Please spell out RHC. This is not a common acronym for readers from different specialties.  

We agree with this and have incorporated your suggestion throughout the manuscript.

5) Is the article also quoted under the references section?

Thank you for pointing this out. However, the article is not quoted under the references section.

6) The legend for Figure 1 should be more informative. Do not expect the readers to interpret it themselves.  

We are grateful to the reviewers for their insightful comment on our paper. We have highlighted the changes of the legend for Figure 1 within the manuscript.

7) This is trivial, but was the written consent obtained from the patient as claimed, or from the family?

Thank you for pointing this out. An informed consent obtained from the closest relative of the patient is attached.

Sincerely Yours,

Elena Angeloudi

Round 2

Reviewer 2 Report

Comments and Suggestions for Authors

The authors have answered most of my concerns, but others must be clarified.

I'm afraid I have to disagree with the answer, “We considered that the phase of the cardiac cycle (diastolic phase) could be interpreted by the fact that the mitral valve was closed.” If I am not wrong, a closed mitral valve occurs during the systolic phase. If the mitral valve is closed, how can blood flow from LA to LV during the diastolic phase? The explanation “We can only speculate that due to the decreased movement of the apical cardiac segments, a negative pressure gradient was generated into the pericardial sack, which caused the translocation of the pericardial effusion to the anti-gravity side of the heart.” As even negative pressure is generated in the pericardial sac, the effusion should still be in the dependent part. Could a Tb pericardial effusion with adhesion cause the loculated effusion? You can use the “crop” function in PowerPoint to arrange EKG images (use TIFF format) to arrange them into one figure.

Author Response

Comment: "I'm afraid I have to disagree with the answer, “We considered that the phase of the cardiac cycle (diastolic phase) could be interpreted by the fact that the mitral valve was closed.” If I am not wrong, a closed mitral valve occurs during the systolic phase. If the mitral valve is closed, how can blood flow from LA to LV during the diastolic phase? The explanation “We can only speculate that due to the decreased movement of the apical cardiac segments, a negative pressure gradient was generated into the pericardial sack, which caused the translocation of the pericardial effusion to the anti-gravity side of the heart.” As even negative pressure is generated in the pericardial sac, the effusion should still be in the dependent part. Could a Tb pericardial effusion with adhesion cause the loculated effusion? You can use the “crop” function in PowerPoint to arrange EKG images (use TIFF format) to arrange them into one figure."

Reply: Thank you for your correction. It is a fact that mitral valve is closed during the systolic phase of the cardiac cycle. We have inadvertently stated the opposite. We have made the corresponding corrention in our revised manuscript. Unfortunately, we have no available images of the pericardial effusion in the diastolic cardiac phase, however, as you mentioned previously, the quantity of the effusion was not clinically concerning in order to perform a pericardiocentesis.

Regarding your comment about the localization of the pericardial effusion, it is true that a Tb pericardial effusion with adhesion could have caused the loculated effusion, something that we have added in our revised manuscript. However, as we have already explained in our original submission,  the patient's lung disease improved considerably after treatment initiation, and therefore we consider it highly unlikely that she developed a new Tb localization, afterwards, in another organ, such as the heart.

As far as the image arrangement, we had already to do that and we found it impossible to fit 12 ECGs with 12 leads each in one figure, since no ECG waveform was comprehensible.

We hope that we have clarified all your inquaries.

Sincerely Yours,

Elena Angeloudi

Round 3

Reviewer 2 Report

Comments and Suggestions for Authors

No more comments

Author Response

Dear Reviewer,

Thank you so much. We are available for any further information.